# Colorless Polyimides Derived from an Alicyclic Tetracarboxylic Dianhydride, CpODA

**DOI:** 10.3390/polym13162824

**Published:** 2021-08-22

**Authors:** Hiroki Ozawa, Eriko Ishiguro, Yuri Kyoya, Yasuaki Kikuchi, Toshihiko Matsumoto

**Affiliations:** 1Department of Industrial Chemistry, Graduate School of Engineering, Tokyo Polytechnic University, Atsugi, Kanagawa 243-0297, Japan; matsumoto9384@gmail.com (H.O.); matsumoto9384@ezweb.ne.jp (E.I.); 2National Institute of Technology, Hachinohe College, Tamonoki, Hachinohe, Aomori 039-1192, Japan; pepino.ouo.0520@gmail.com (Y.K.); kikumal-g@hachinohe-ct.ac.jp (Y.K.)

**Keywords:** CpODA, colorless polyimides, CPI, alicyclic polyimides, polyalicyclic structure, high Tg, low CTE

## Abstract

An alicyclic tetracarboxylic dianhydride having cyclopentanone bis-spironorbornane structure (CpODA) was polycondensated with aromatic dianhydrides to form the corresponding poly(amic acid)s which possessed logarithmic viscosities in the range 1.47–0.54 dL/g. The poly(amic acid) was imidized by three methods: a chemical, a thermal, and a combined chemical and thermal process. In a thermal method, imidization temperature markedly influenced the film quality and molecular weight of the polyimide. When the poly(amic acid) was cured over the Tg of the corresponding polyimide, the flexible polyimide films were obtained and the molecular weights increased several times, which means that the post-polymerization took place. In spite of low-temperature cure below Tg flexible films with the imidization ratio of 100% were fabricated by a combined chemical and thermal imidization technique. The films possessed the decomposition temperatures in a range of 475–501 °C and Tgs over 330 °C. The high Tg results from a dipole–dipole interaction between the keto groups of the polymer chains as well as development of the rigid polyalicyclic unit. The polyimide films exhibited CTE between 17 and 57 ppm/K. All the films fabricated were entirely colorless and possessed the λcut-offs shorter than 337 nm. Notably, the films prepared by a chemical method exhibited outstanding optical properties.

## 1. Introduction

The earliest of the thermally stable polymers, and still now one of the most common commercial materials, is an aromatic polyimide that was synthesized by DuPont in 1964 and marketed under the name H-Film and later Kapton^®^ (E.I. du Pont Co.) [1]. It exhibits not only excellent electrical and high-temperature mechanical properties but also cosmic ray resistance [2]. Therefore, aromatic polyimides are used as protective coatings of space flyer unit, and insulation coatings of motor wire for electric vehicles and trains, and microelectronic components such as circuit boards, flexible cable, and insulators. However, aromatic polyimides become reddish yellow due to the intra- and inter-molecular charge transfer [3]. Consequently, they cannot be used in areas where colorlessness and transparency are important requirements. For the first time, Bikson et al. directed their attention to reasons for discoloration of aromatic polyimides like Kapton^®^ [4]. Dine-Hart et al. reported that the discoloration was due to CT complex formation using the PI model compound [5] and Gordina et al. reached a similar conclusion using polyimide itself [6]. We investigated the origin of discoloration of Kapton^®^-type polyimide from the viewpoint of quantum chemistry using the model compound, and reported that the light absorption is attributable to intramolecular CT from diamine moiety to dianhydride one [7]. 

The first example of a colorless polyimide prepared from a hexafluoroisopropylidene-bridged phenylene dianhydride (6FDA) and a diamine was reported as a patent by Rogers (du Pont de Nemours, E. I., and Co.) more than half a century ago [8]. St. Clair et al. demonstrated that polyimides containing the hexafluoroisopropylidene and/or sulfone linkage exhibited high transparency in the visible region. Their approach was to separate the chromophoric groups and reduce the electronic interaction between the color-causing centers in the polymer molecular structure by employing the steric hindrance and electronic effects of the groups [9]. At the present, the effective methods to develop colorless polyimide films mainly include introducing groups with high electronegativity characteristics, such as trifluoromethyl groups, sulphone groups, or substituents with non-conjugated characteristics, such as aliphatic or alicyclic groups, and groups with large molar volume like fluorene or cardo groups into the PI molecular chains [10,11,12,13,14,15,16,17,18,19].

Recently, colorless polyimide films have been paid much attention due to their excellent combined thermal, optical, and dielectric properties. The colorless polyimide films are expected to be a promising candidate for advanced optoelectronic materials, such as substrates for flexible display devices [20,21,22], flexible organic solar cells [23], and flexible heater for wearable sensors [24], encapsulants for displays or electronic chips [25], and other high-tech applications [26,27]. The introduction of polyalicyclic unit into polyimide backbone would facilitate less polymer–polymer interaction and enhance the solubility in organic solvents. The lack of color is generally associated with the inhibition of CT interactions. The high-temperature stability of polyalicyclic polyimides can be explained by the diamond-like multibond structure. Less probability of main chain scission fosters the high-temperature stability, and multibond structure increases the main chain rigidity and enhances the glass transition temperature (Tg) [28,29].

In the present work, as one of our continuous studies developing high-performance colorless polyimide films for advanced optical applications, the enhancement of thermal and optical properties was endeavored not only by introducing a cyclopentanone bis-spironorbornane unit into the molecular structure but also by adopting a low temperature film-fabrication method. The CpODA-based colorless polyimides prepared in the present study exhibited the much higher thermal stability than those previously reported. The effects of the introduced unit and the fabrication method on the optical and thermal properties of the derived colorless polyimide films will be described in detail.

## 2. Materials and Methods

### 2.1. Materials

Cyclopentanone bis-spironorbornane tetracarboxylic dianhydride (CpODA) was synthesized according to our previous literature [30] or was donated from ENEOS Co. Ltd. (Tokyo, Japan), and dried at 170 °C under vacuum for 3 h prior to use. Aromatic diamines were supplied from Wakayama Seika Kogyo Co., Ltd. (Wakayama, Japan) and were purified using a glass tube oven (Shibata GTO-2000) by vacuum distillation just prior to use. *N*,*N*-Dimethylacetamide (DMAc) was purchased from FUJIFILM Wako Pure Chemical Co. (Osaka, Japan), and used without further purification. The other reagents were used as received.

### 2.2. Measurements

The viscosities of poly(amic acid) (PAA) solutions at a concentration of 0.5 g/dL in DMAc were measured using an Ostwald viscometer at 30 °C. The molecular weight was analyzed using a JASCO 2080 SEC equipped with an RI detector and Shodex KF-806M columns at 40 °C (development solvent, chloroform). Molecular weight calculations were made on the basis of polystyrene standards. The imidization ratio of PI(CpODA+3,4′-DDE) was estimated by ^1^H-NMR spectroscopy according to our previous paper [31]. The ^1^H-NMR spectra were obtained using a JEOL JNM-LA500 spectrometer and deuterochloroform (chloroform-d) was used as the solvent. Thermal analyses were carried out using a Seiko SSC 5200-TG/DTA 220 instrument at a heating rate 10 K/min under nitrogen with a flow rate of 200 mL/min for the thermogravimetric analysis (TGA). The glass transition temperatures were determined using a Seiko TMA/SS100 thermomechanical analyzer equipped with a penetration probe of 1 mm diameter with an applied constant load of 10 g (stress, 0.125 MPa) at a heating rate of 10 K/min in air. The coefficient of thermal expansion (CTE) of the polyimide film (10 mm long, 3.5 mm wide, and 10–15 μm thick) was measured in a temperature range of 40–240 °C at a heating rate of 5 K/min on a Seiko TMA/SS100 instrument with a drawing load of 0.35 g in μm, namely 980 mN/mm^2^, in a slow stream of nitrogen. The CTE value was evaluated from the averaged slope in the range of 100–200 °C of TMA curve during second run. The Infrared spectra were recorded using a JASCO 460 Plus Fourier transform spectrophotometer. UV-vis spectra of the free-standing polyimide films were recorded on a JASCO V-570 UV/vis/NIR spectrophotometer. The thickness and the refractive index/birefringence of polyimide films were measured using a Metricon Model 2010/M prism coupler.

### 2.3. General Polymerization Procedure for Poly(Amic Acid) Preparation

In a 30-mL three-necked flask equipped with a mechanical stirrer were placed 4,4′-DDE (0.4005 g, 2.000 mmol) and 2.68 g of DMAc. As a slow stream of dry nitrogen gas was maintained, the mixture was stirred until the diamine was entirely dissolved. The dianhydride CpODA (0.7688 g, 2.000 mmol) and an additional 2.00 g of DMAc were added, and the mixture was stirred at a rate of 50 rpm for 1 day at room temperature.

### 2.4. Imidization and Film Fabrication

#### 2.4.1. Thermal Imidization Method

An aliquot of the polymerization solution containing PAA was cast on a glass plate using a doctor blade. The polyimide film was fabricated by heating the plate at 80 °C for 2 h, then 350 °C for 0.5 h under vacuum. The glass plate was immersed into boiling water to remove the polyimide film.

#### 2.4.2. Chemical Imidization Method

A mixture of acetic anhydride (6.00 mmol) and *N*-methylpiperidine (2.0 mmol) was added to the poly(amic acid) solution in one portion, and the mixture was heated at 70 °C. After 30 min the white solid appeared, and the suspension was poured into methanol. The solid was collected by filtration and dried at room temperature under vacuum. The powdery polyimide was dissolved in chloroform at a solid content of 10 wt %. The solution was cast on three glass plates. One plate was heated at 80 °C for 2 h and the others were cured under vacuum at 200 °C and at 350 °C, respectively. The glass plate was immersed into boiling water to assist the film in peeling off.

#### 2.4.3. Combined Chemical and Thermal Imidization Method

This method was done according to our previous paper [31]. Triethylamine (TEA, 1.0 mmol) and trifluoroacetic anhydride (TFAA, 0.6 mmol) were added to the PAA solution, and the mixture was stirred magnetically for 18 h at room temperature. The solution was cast on a glass plate, and the plate was heated at 80 °C for 2 h, then at the prescribed temperature for 1 h under vacuum.

## 3. Results and Discussion

### 3.1. Polymer Synthesis and Film Fabrication

Polyimides were synthesized by a two-step method where the first step includes the PAA formation at room temperature in DMAc (Scheme 1). Figure 1 shows the variation in the logarithmic viscosity of the PAA solution with elapsed time since adding CpODA into the 4,4′-DDE solution. The dianhydride CpODA was hardly soluble in DMAc and it took about 6 h to dissolve completely in the 4,4′-DDE solution. The viscosity measurement was carried out after entire dissolution of CpODA. The viscosity increases with an increase of the time, and maximum is closely approached after about 14 h. The viscosity is related with the weight-averaged molecular weight (Mw) and both the values increase as polymerization proceeds. After the Mw reaches to the maximum, it decreases gradually due to amide exchange reaction between high and low molecular weight PAA chains, which results in the viscosity curve exhibiting maximum peak around 14 h. The PAA solutions prepared from CpODA and aromatic diamines possessed the logarithmic viscosities in a range of 1.47–0.54 dL/g.

In the second step, the PAA was imidized through three methods involving a chemical, a thermal or a combined chemical and thermal process using the same lot of PAA. In a conventional thermal method, imidization temperature influenced dramatically the film quality and molecular weight of the polyimide (Table 1, Figure 2). When PAA(CpODA+1,3-BAB) prepared from CpODA and 1,3-bis(4-aminophenoxy)benzene was cured at 250 °C after being cast on the glass plate, the resulting PI(CpODA+1,3-BAB) film was brittle and the number-averaged molecular weight (Mn) was 41 thousands. However, on curing the PAA over the glass transition temperature of the PI(CpODA+1,3-BAB) (Tg = 290 °C), the flexible polyimide films were obtained, and the molecular weights increased several times. In the case of PAA(CpODA+3,4′-DDE), the flexible film was given only by curing at 350 °C. The molecular weights increased 5–8 times by curing around or over the Tg, which means that the post-polymerization took place at the temperature [30]. An imaginary depiction of post-polymerization around Tg is shown in Figure 3. The PAAs undergo depolymerization at intermediate-temperature region (100–250 °C), which results in the molecular weight decrease. As increasing temperature up to 350 °C, post-polymerization occurs around the Tg in solid state and the polymer recovers high molecular weight. It is known that similar phenomenon happens in aromatic polyimide synthesis [32].

When the colorless polyimide (CPI) film was cured at high temperature, the transparency was decreased with an increase in curing time and temperature. Prolonged curing time reduced the averaged transparency of 400–800 nm by 6% after 10-h cure at 250 °C in air. In the case of chemically imidized film, only 1.5% reduction of the averaged transparency was observed. The difference can be explained by end-capping effect of the amino terminal of the polyimide chain [33]. A conventional chemical imidization was carried out by adding a mixture of acetic anhydride and a base to the PAA solution, and then the mixture was heated at 70 °C for two hours. However, this method cannot be used because most CpODA-based polyimides were precipitated in the solvent during chemical imidization reaction. High Tg polymers like full-imidized polyimides are generally insoluble in organic solvents due to the rigid structure and a strong interaction between polymer chains. Recently, we have developed a combined chemical and thermal imidization method (Scheme 2), that is, in the first step, the PAA was partially, about 30%, imidized using a chemical imidization technique at room temperature for 16 h, which gave a homogeneous solution [31]. In the second step, the solution was cast, then cured at 200 °C to give a flexible film. In spite of low-temperature cure below Tg, the imidization ratio reached to 100%.

The SEC profiles of polyimide films prepared from CpODA and 3,4′-DDE using a thermal (“T”) and combined chemical and thermal (“C + T”) methods at each 200 °C or 350 °C cure are shown in Figure 4a. The number-averaged (Mn) and the weight-averaged (Mn) molecular weights are plotted as a function of the cured temperatures in Figure 4b, where blue dashed lines and red solid lines denote the polyimides prepared by “T” and “C + T” methods, respectively. In each method, imidization temperature markedly influences the molecular weight of the polyimide film. When the PAA was cured at 200 °C in the case of “T”, the film was brittle. However, on curing the PAA at 350 °C, the flexible polyimide film was obtained and the molecular weight increased several times, as previously described. Similarly, in the combined method, high temperature cure of the second step enhances the molecular weight, as the partial imidization (ca. 30%) suppresses the degradation of PAA and the regenerated amino groups can undergo the post-polymerization with anhydride moiety at high temperature.

The SEC profiles of the PI(CpODA+3,4′-DDE) films prepared by the chemical imidization method are shown in Figure 5, where blue, orange, and red line curves denote the PI films cured at 80, 200, and 350 °C, respectively. Three curves have almost the same shape and maximum peak position. This means that high molecular weight is fixed due to suppression or inhibition of depolymerization of PAA by a chemical imidization. Furthermore, post-polymerization was not undergone due to the absence of free amino group. 

### 3.2. Properties of the Polyimide Films

#### 3.2.1. Thermal Properties

The thermal properties of the polyimide films prepared by three different methods are summarized in Table 2. The films possess the 5% weight-loss temperatures (T5) over 450 °C and the decomposition temperatures (Td) in a range of 475–501 °C. Td is noted as the intersection of the extrapolations of the two slopes in the TGA curve. Introduction of polyalicyclic unit results in less probability of the main chain scission at high temperature. The polyimides have Tgs over 330 °C except for 1,3-BAB-based one, which consists of the diamine having two kinked ether linkages. The high Tg would be most likely due to a dipole–dipole interaction between the keto groups of the polymer chains (Figure 6) and also be attributable to development of the rigid polyalicyclic unit. The polyimide films exhibit CTE between 17 and 57 ppm/K. Comparing the CTE of PI(CpODA+3,4′-DDE) films, the value of chemically imidized film is lower than that of the thermally imidized one. In plain-oriented polymer chain, orientation relaxation might occur at high temperature in the curing process. In other words, non-orientation relaxation might take place in the chemical imidization process [34].

#### 3.2.2. Optical properties

The UV-vis transmission spectra of the polyimide films prepared from CpODA and aromatic diamines using three different methods are displayed in Figure 7. All the films fabricated in this study are highly transparent and entirely colorless. The transparency averaged in the visible region (Tvis) and the cut-off wavelength (λcut-off), where the transmittance becomes 1%, are listed in Table 3. The polyimide films possess the λcut-offs shorter than 337 nm, and the Tvis values of the films prepared by a chemical (C) and a combined imidization (C + T) methods are higher than that of the thermally imidized one. Especially, the films prepared by a chemical method exhibited outstanding optical properties, although the films were post-cured at 350 °C in N_2_ after imidization. It is ascribable to an “end-capping effect” where an amino group attached at polymer ends, easily oxidized and colored, were protected by acylation [33].

The thickness and refractive index at three points of the polyimide film were measured using a Metricon Model 2010/M Prism Coupler [35]. The in-plane and out-of-plane retardations (R0 and Rth) of the polyimide film are given by the following equations, where nx, ny, and nz are refractive indexes of x, y, and z axis directions, and *d* denotes thickness of the film in nm unit (Figure 8a).
R0 = (nx − ny) × *d*
Rth = [(nx + ny)/2 − nz] × *d*

The retardation values normalized by being divided with the film thickness were estimated as a 1 μm thick film and are list in Table 4 together with the in-plane CTE. The CTE of the thermal imidized PI(CpODA+4,4′-DABA) (T) is slightly lower than that of the film fabricated by a combined C + T method. The 4,4′-DABA diamine contains an amide-linkage with a planar structure which gives a rod-like nature to the polyimide backbone. Low-CTE polyimides exhibit a large Rth value, which means that the polyimide chains align mainly in-plane direction. The orientation reduces the in-plane CTE, and the film might be expanded toward the out-of-plane direction, assuming that the volume coefficients of thermal expansion of these amorphous polyimides are almost the same (Figure 8b).

## 4. Conclusions

Cyclopentanone bis-spironorbornane tetracarboxylic dianhydride CpODA was polycondensated with aromatic dianhydrides in DMAc at room temperature to form the corresponding poly(amic acid)s which possessed logarithmic viscosities in the range 1.47–0.54 dL/g. The poly(amic acid) was imidized by three methods involving a chemical, a thermal, or a combined chemical, and a thermal process using the same lot of poly(amic acid). In a thermal method, imidization temperature dramatically influenced the film quality and molecular weight of the polyimide. On curing the poly(amic acid) over the glass transition temperature of the corresponding polyimide, the flexible polyimide films were obtained and the molecular weights increased several times, which means that the post-polymerization took place around the Tg. In spite of low-temperature cure below the Tg flexible films with the imidization ratio of 100% were fabricated by a combined chemical and thermal imidization method. The films possessed the 5% weight-loss temperatures over 450 °C and the decomposition temperatures in a range of 475–501 °C. The polyimides had Tgs over 330 °C, except for 1,3-BAB-based one. The high Tg results from a dipole–dipole interaction between the keto groups of the polymer chains, as well as development of the rigid polyalicyclic unit. The polyimide films exhibited CTE between 17 and 57 ppm/K. All the films fabricated in this study were entirely colorless and possessed the λcut-offs shorter than 337 nm. Notably, the films prepared by a chemical method exhibited outstanding optical properties due to an “end-capping effect”.

The CpODA-based polyimide films have exceptional thermal stability and outstanding transparency, making them a promising candidate of the flexible substrate to respond to the growing demand IT systems, such as foldable smart phone or tablet computers.

## Data Availability

The data presented in this study are available on request from the corresponding author.

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
