# Peer review of "Colorless Polyimides Derived from an Alicyclic Tetracarboxylic Dianhydride, CpODA"

_polymers, 2021, doi:10.3390/polym13162824_

Round 1

Reviewer 1 Report

This is a review report for the paper titled “Colorless polyimides derived from an alicyclic tetracarboxylic dianhydride, CpODA” submitted to Polymers by Matsumoto et al. In this paper, the authors reported soluble and colorless polyimides based on bis-spironorbornane tetracarboxylic dianhydride, CpODA, and aromatic diamines and the relation between molecular weight and imidization methods; conventional thermal, chemical, and chemical-thermal combined methods, as well as the thermal stability, thermal expansion, and optical properties; absorption of UV-visible region and birefringence. Recently, as the authors stated, colorless polyimides have got much more attention for applications as transparent flexible substrates with high thermal stability. The authors discussed the molecular weight on the basis of quantitative SEC measurements and revealed that higher imidization temperature than Tg of the PI resulted in higher molecular weight leading to the flexibility of the films, while lower curing temperature resulted in lower molecular weight and fragile films. Researchers interested in colorless PIs are increasing recently and such a detailed report is very useful for such researchers. However, several minor points listed below are better to be modified before accepted for Polymers.

  1. With respect to Fig. 1, the viscosity of the solution of the PAA of CpODA and 4,4′-DDE exhibited maximum value at around 14 h. Some conceivable reason is better to be provided. Is it possible for the amide bonds in the PAA to exchange between higher and lower molecular weight chains?
  1. I didn’t find the value of the imidization ratio of the PI films compared in Fig. 2, 4, 5, and 6, although the authors stated that the “C+T” methods leaded to 100% imidization ratio even at low curing temperature of 200°C. If possible, please present FTIR spectra of the PI films and imidization ratio or TGA curves.
  1. In page 4, line 159-160, the authors stated “It is known that similar phenomenon happens in aro- matic polyimide synthesis”. However, related papers were not cited in this part. If possible, please add appropriate literatures. Or, it is the statement based on the authors’ unpublished experiments so far?
  1. Fig. 4(a) and Fig. 6 looked almost same and I think these figures can be combined.
  1. It is trivial but I couldn’t grasp the meaning of the sentence in page 3, line 100; “The value was evaluated as an average with 100-200 oC for the film plane direction at the second run”. Please reconsider it. I supposed “The CTE value was evaluated from the averaged slope in the range of 100–200°C of TMA curve during second run.

Author Response

 Thank you for your suggestive and valuable comments. Please see the attachment.

Reviewer 2 Report

Recommendation: Minor Revision 

This manuscript proposed an alicyclic tetracarboxylic dianhydride that has cyclopentanone bis-spironorbornane structure (CpODA) polycondensated with aromatic dianhydrides and formed poly(amic acid)s with different diamines. The poly(amic acid) was imidized by three methods to fabricate colorless polyimide films.  and the optical and thermal properties of the films were investigated. The results show that the films fabricated by a combined  chemical and thermal imidization technique possessed the decomposition temperatures in a range of 475-501 ℃ and Tgs over 330 ℃ and the CTE between 17 and 57 ppm/K. All the films fabricated were colorless and have the λcut-offs shorter than 337 nm. I would say, these results are of interest and deserve publication in Macromolecular Chemistry and Physics to attract the attention of the readership after the authors consider the following.

  1. The authors would better conduct some compassion between the fabricated films with current almost commercial CPI such as Kapton films to demonstrate their standout merits.
  1. At a temperature lower than Tg, colorless polyimide films were fragile and had a low molecular weight (Table 1). The author would better measure the infrared spectra of polyimide films at different temperatures to determine the degree of imidization. At the same time, the molecular weight of PAA before imidization should be detected and compared with that after films formation.
  1. It is recommended that Figure 3 be reordered from top to bottom in order of reaction processes. In addition, the diagrams of depolymerization and recombination processes should not be exactly the same. As for the problem of depolymerization and recombination in Figure 3, if depolymerization occurs at 100-250℃ as stated by the author, does this process take place from the beginning of drying at room temperature? If so, what caused it. If not, what causes it to suddenly depolymerize during dehydration.
  1. The authors have failed to specify where the data came from. Such as “Prolonged curing time reduced the averaged transparency of 400-800 nm by 6% after 10-hour cure at 250 ℃ in air. In the case of chemically imidized film, only 1.5 % reduction of the averaged transparency was observed ”(Line 188-190,page 6), and “the imidization ratio reached to 100 %”(Line 203, page 6). It is better for the authors to list the specific experimental test methods and results in the article. Meanwhile, Figure 5 and Figure 6 should have the same values on the abscissa.(Line 245 and Line 250)

Author Response

The authors would like to appreciate giving suggestive and valuable comments. Please see the attachment.
